# Precision Gating: Improving Neural Network Efficiency with Dynamic Dual-Precision Activations

**Yichi Zhang**
Cornell University
yz2499@cornell.edu

**Ritchie Zhao**
Cornell University
rz252@cornell.edu

**Weizhe Hua**
Cornell University
wh399@cornell.edu

**Nayun Xu**
Cornell Tech
nx38@cornell.edu

**G. Edward Suh**
Cornell University
edward.suh@cornell.edu

**Zhiru Zhang**
Cornell University
zhiruz@cornell.edu

## Abstract

We propose precision gating (PG), an end-to-end trainable *dynamic dual-precision quantization* technique for deep neural networks. PG computes most features in a low precision and only a small proportion of important features in a higher precision to preserve accuracy. The proposed approach is applicable to a variety of DNN architectures and significantly reduces the computational cost of DNN execution with almost no accuracy loss. Our experiments indicate that PG achieves excellent results on CNNs, including statically compressed mobile-friendly networks such as ShuffleNet. Compared to the state-of-the-art prediction-based quantization schemes, PG achieves the same or higher accuracy with 2.4× less compute on ImageNet. PG furthermore applies to RNNs. Compared to 8-bit uniform quantization, PG obtains a 1.2% improvement in perplexity per word with 2.7× computational cost reduction on LSTM on the Penn Tree Bank dataset.

## 1 Introduction

In recent years, deep neural networks (DNNs) have demonstrated excellent performance on many computer vision and language modeling tasks such as image classification, semantic segmentation, face recognition, machine translation, and image captioning (Krizhevsky et al., 2012; He et al., 2016a; Ronneberger et al., 2015; Chen et al., 2016; Zhao et al., 2018; Schroff et al., 2015; Luong et al., 2015; Vaswani et al., 2017). One evident trend in DNN design is that as researchers strive for better accuracy, both the model size and the number of DNN layers have drastically increased over time (Xu et al., 2018). At the same time, there is a growing demand to deploy deep learning technology in edge devices such as mobile phones, VR/AR glasses, and drones (Wu et al., 2019). The limited computational, memory, and energy budgets on these devices impose major challenges for the deployment of large DNN models at the edge.

DNN quantization is an important technique for improving the hardware efficiency of DNN execution (Zhao et al., 2017). Numerous studies have shown that full-precision floating-point computation is not necessary for DNN inference — quantized fixed-point models produce competitive results with a small or zero loss in prediction accuracy (Lin et al., 2016; He et al., 2016b; Zhou et al., 2016; 2017). In some cases, quantization may even improve model generalization by acting as a form of regularization. Existing studies mainly focus on static quantization, in which the precision of each weight and activation is fixed prior to inference (Hubara et al., 2017; He et al., 2016b). Along this line of work, researchers have explored tuning the bitwidth per layer (Wu et al., 2018b; Wang et al., 2019; Dong et al., 2019) as well as various types of quantization functions (Wang et al., 2018; Courbariaux et al., 2016; Li et al., 2016; Zhou et al., 2016). However, static DNN quantization methods cannot exploit input-dependent characteristics, where certain features can be computed in a lower precision during inference as they contribute less to the classification result for the given input. For

example, in computer vision tasks, the pixels representing the object of interest are typically more important than the background pixels.

In this paper, we reduce the inefficiency of a statically quantized DNN via *precision gating* (PG), which computes most features with low-precision arithmetic operations and only updates few important features to a high precision. More concretely, PG first executes a DNN layer in a low precision and identifies the output features with large magnitude as important features. It then computes a sparse update to increase the precision of those important output features. Intuitively, small values make a small contribution to the DNN's output; thus approximating them in a low precision is reasonable. Precision gating enables dual-precision DNN execution at the granularity of each individual output feature, and therefore greatly reducing the average bitwidth and computational cost of the DNN. We further introduce a differentiable gating function which makes PG applicable to a rich variety of network models.

Experimental results show that PG achieves significant compute reduction and accuracy improvement on both CNNs and LSTMs. Compared to the baseline CNN counterparts, PG obtains 3.5% and 0.6% higher classification accuracy with up to $4.5\times$ and $2.4\times$ less computational cost for CIFAR-10 and ImageNet, respectively. On LSTM, compared to 8-bit uniform quantization PG boosts perplexity per word (PPW) by 1.2% with $2.8\times$ less compute on the Penn Tree Bank (PTB) dataset. Our contributions are as follows:

1. We propose precision gating (PG), which to the best of our knowledge is the first end-to-end trainable method that enables dual-precision execution of DNNs. PG is applicable to a wide variety of CNN and LSTM models.
2. PG enables DNN computation with lower average bitwidth than other state-of-the-art quantization methods. By employing a low-cost gating scheme, PG has the potential to reduce DNN execution costs in both commodity and dedicated hardware.
3. PG achieves the same sparsity during back-propagation as forward propagation, which dramatically reduces the computational cost for both passes. This is in stark contrast to prior dynamic DNN optimization methods that focus only on reducing the inference cost.

## 2 RELATED WORK

**Quantizing activations.** Prior studies show that weights can be quantized to low bitwidth without compromising much accuracy (Zhu et al., 2017); however, quantizing activations with a low bitwidth (e.g., 4 bits) typically incurs a nontrivial accuracy degradation (Mishra et al., 2018; Zhou et al., 2016). This is partially caused by large activation and weight outliers, which stretch the quantization grid too wide and too sparse under a low precision, thus increasing the error (Park et al., 2018; Zhao et al., 2019). To address this problem, Choi et al. (2018) propose PACT to reduce the dynamic range of the activations through clipping the outliers using a learnable threshold. PACT provides a more effective quantization scheme under a very low bitwidth (e.g., 4 bits). In this work we incorporate the PACT method in the training flow of PG to handle the large outliers.

**Prediction-based execution.** Prior works have explored predicting ReLU-induced zeros and max-pooling compute redundancy to lower the computational cost of CNNs. For example, Lin et al. (2017); Song et al. (2018) propose *zero-prediction* to utilize a few of the most-significant bits in the input activations to predict the sign of the output activation. Zero-prediction removes redundancy by exploiting the fact that negative outputs will be suppressed by the ReLU anyway. Yet, this method only applies to ReLU activations and only when a linear layer is directly followed by ReLU. Hence such method does not apply to RNNs that use sigmoid or tanh as the activation function and many modern CNN networks that use batch normalization (Ioffe & Szegedy, 2015) before ReLU. Hua et al. (2019a;b) propose channel gating to dynamically turn off a subset of channels that contribute little to the model prediction result. Precision gating is orthogonal to this pruning technique as channel gating executes the whole network at the same precision.

More recently, Cao et al. (2019) propose *SeerNet*, which also executes a CNN model in dual precision. For each convolutional layer, SeerNet first executes a quantized version of the layer and uses the results to predict the output sparsity induced by the ReLU or the computational sparsity induced by the max-pooling layer. For those activations that are not suppressed according to the prediction, SeerNet computes the original convolution in full-precision (32-bit float). One key dif-

Input Activations $\mathbf{I} = \mathbf{I}_{hb} << B_{lb} + \mathbf{I}_{lb}$

Figure 1: Splitting an input feature $\mathbf{I}$ into $\mathbf{I}_{hb}$ (blue), the most-significant $B_{hb}$ bits, and $\mathbf{I}_{lb}$ (orange), the remaining $B_{lb}$ bits. The total bitwidth is $B$.

ference between PG and SeerNet is that PG reuses the result of the low-precision compute as a partial product when it performs the high-precision multiplication in the update phase. In contrast, the full-precision compute in SeerNet does not reuse the output from the quantized layer, which incurs a higher execution cost.

**Feature-level precision tuning.** There is also prior work that uses a different precision to handle the outlier quantization. Park et al. (2018) propose value-aware quantization where the majority of data are computed at reduced precision while a small number of outliers are handled at high precision. Our approach is significantly different because we allow dual precision for every feature, not only for the outliers.

## 3    PRECISION GATING (PG)

In this section we first describe the basic mechanism of PG. We then discuss how to design the gating scheme to accelerate both forward and backward passes. Finally, we consider incorporating outlier clipping to reduce the quantization error.

### 3.1    BASIC FORMULATION

We first define a linear layer in a neural network (either convolutional or fully-connected) as $\mathbf{O} = \mathbf{I} * \mathbf{W}$, where $\mathbf{O}$, $\mathbf{I}$, and $\mathbf{W}$ are the output, input, and weights, respectively. Suppose $\mathbf{I}$ is represented in a $B$-bit fixed-point format, which is shown in Figure 1. PG partitions $\mathbf{I}$ into (1) $\mathbf{I}_{hb}$, the $B_{hb}$ most-significant bits (MSBs), and (2) $\mathbf{I}_{lb}$, the remaining $B_{lb}$ least-significant bits (LSBs). Here $B = B_{hb} + B_{lb}$. More formally, we can write:

$$\mathbf{I} = \mathbf{I}_{hb} << B_{lb} + \mathbf{I}_{lb} \qquad (1)$$

Here $<<$ denotes the left shift operator. We can then reformulate a single $B$-bit linear layer into two lower-precision computations as:

$$\mathbf{O} = \mathbf{O}_{hb} + \mathbf{O}_{lb} = [\mathbf{W} * (\mathbf{I}_{hb} << B_{lb})] + [\mathbf{W} * \mathbf{I}_{lb}] \qquad (2)$$

$\mathbf{O}_{hb}$ is the partial product obtained by using the MSBs of input feature ($\mathbf{I}_{hb}$) whereas $\mathbf{O}_{lb}$ represents the remaining partial product computed with $\mathbf{I}_{lb}$.

Precision gating works in two phases. In the **prediction phase**, PG performs the computation $\mathbf{O}_{hb} = \mathbf{W} * (\mathbf{I}_{hb} << B_{lb})$. Output features of $\mathbf{O}_{hb}$ greater than a learnable gating threshold $\Delta$ are considered important. In the **update phase**, PG computes $\mathbf{O}_{lb} = \mathbf{W} * \mathbf{I}_{lb}$ only for the important features and adds it to $\mathbf{O}_{hb}$. The overall execution flow of PG is illustrated in Figure 2, where unimportant output features only take the upper path while important ones are computed as the sum of both paths. More precisely, this can be summarized as follows:

$$\mathbf{O} = \begin{cases} \mathbf{O}_{hb} & \mathbf{O}_{hb} \leq \Delta \\ \mathbf{O}_{hb} + \mathbf{O}_{lb} & \mathbf{O}_{hb} > \Delta \end{cases} \qquad (3)$$

In essence, PG intends to compute the majority of (unimportant) features with $B_{hb}$ bits and only a small set of important features with $B$ bits. The importance of each element in the output $\mathbf{O}$ is calculated by comparing the magnitude of its partial sum $\mathbf{O}_{hb}$ to $\Delta$. Let $Sp$ be the percentage of

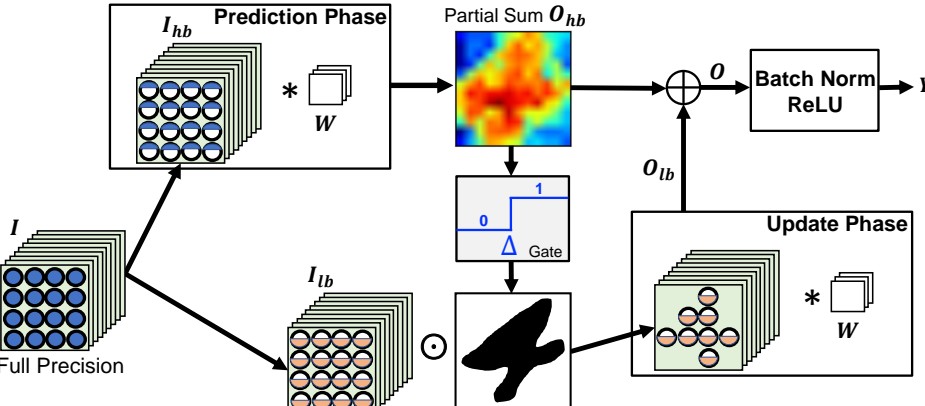

Figure 2: **The PG building block in CNN models –** Input features are split into $\mathbf{I}_{hb}$ and $\mathbf{I}_{lb}$. In the prediction phase, $\mathbf{I}_{hb}$ first convolves with the full precision filters $\mathbf{W}$ to obtain $\mathbf{O}_{hb}$. In the update phase, if the partial sum $\mathbf{O}_{hb}$ of a feature exceeds the learnable threshold $\Delta$, we will update that feature to high-precision by adding $\mathbf{O}_{lb}$ to $\mathbf{O}_{hb}$. Otherwise, we skip the update phase, and the output feature therefore remains computed at ultra low-precision. The prediction and update phases share the same weights. $\odot$ denotes the Hadamard product.

unimportant activations over all features. PG saves $\frac{Sp \cdot B_{lb}}{B}$ fraction of the compute in the original DNN model. PG thus achieves dual-precision execution using a lightweight gating mechanism which only adds a comparison operation. The goal is to minimize the *average bitwidth* of the multiply-add operations in the DNN execution.

## 3.2 EFFICIENT LEARNABLE GATING SCHEME

PG automatically learns a gating threshold $\Delta_{c,l}$ during training for each output channel in each DNN layer. A larger $\Delta_{c,l}$ indicates that more output features are computed in low-precision, resulting in greater computational savings but possibly at the expense of reduced model accuracy. Define $\Delta$ as the vector containing each gating threshold $\Delta_{c,l}$. We formulate the problem of optimizing $\Delta$ as minimizing the original model loss $L$ along with an L2 penalty term:

$$\min_{\mathbf{W},\Delta} L(\mathbf{I}, y; \mathbf{W}, \Delta) + \sigma \left\| \Delta - \delta \right\|^2 \tag{4}$$

Here $y$ is the ground truth label, $\sigma$ is the penalty factor, and $\delta$ is the *gating target*, a target value for the learnable threshold. The penalty factor and gating target are hyperparameters which allow a user to emphasize high computation savings (large $\sigma$ or $\delta$) or accuracy preservation (small $\sigma$ or $\delta$).

Training a model with precision gating can be performed on commodity GPUs using existing deep learning frameworks. We implement PG on GPU as the equation $\mathbf{O} = \mathbf{O}_{hb} + \mathbf{mask} \odot \mathbf{O}_{lb}$, where $\mathbf{mask} = \mathbf{1}_{\mathbf{O}_{hb} > \Delta}$ is a binary decision mask and $\odot$ represents element-wise multiplication. During forward propagation, most elements in $\mathbf{mask}$ are 0. PG therefore saves hardware execution cost by only computing a sparse $\mathbf{O}_{lb}$ in the update phase. If PG is implemented in a dedicated hardware accelerator, MSBs and LSBs will be wired separately; The prediction phase can be controlled by a multiplexer, and only computes LSB convolutions while $\mathbf{O}_{hb}$ exceeds the threshold, thus achieving savings in both compute cycles and energy.

The $\mathbf{mask}$ is computed using a binary decision function (i.e., a step function), which has a gradient of zero almost everywhere. To let gradients flow through $\mathbf{mask}$ to $\Delta$, we use a sigmoid on the backward pass to approximate the step function. Specifically, we define $\mathbf{mask} = \mathrm{sigmoid}(\alpha(\mathbf{O}_{hb} - \Delta))$ only on the backward pass as in the previous work Hua et al. (2019b). Here $\alpha$ changes the slope of the sigmoid, thus controlling the magnitude of the gradients.

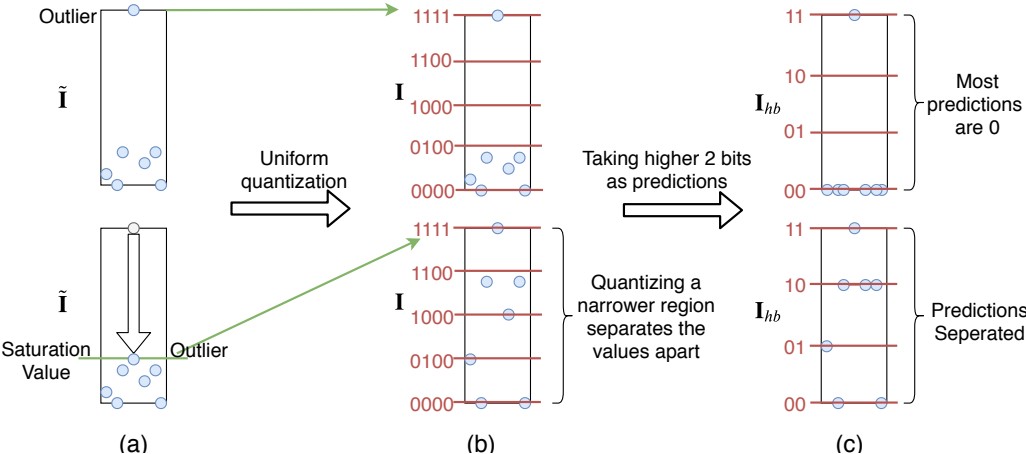

Figure 3: **Effect of clipping** – A toy example illustrating how a clip threshold helps separating prediction values apart. The first row is quantization and prediction without a clip threshold, while the second row has a clip threshold. **(a)** Distribution of floating-point input features $\tilde{\mathbf{I}}$. **(b)** Distribution of $\mathbf{I}$ after quantizing $\tilde{\mathbf{I}}$ to 4 bits. **(c)** Distribution of $\mathbf{I}_{hb}$ which takes the higher 2 bits of $\mathbf{I}$.

### 3.3 SPARSE BACK-PROPAGATION

A sparse update phase only reduces the computational cost during the inference (forward propagation). We further propose to save the compute during the back-propagation by modifying the forward function of the PG block. Specifically, we square the **mask** element-wise.

$$\mathbf{O} = \mathbf{O}_{hb} + \mathbf{mask}^2 \odot \mathbf{O}_{lb} \tag{5}$$

Given that **mask** is a binary tensor, $\mathbf{mask}^2$ in Eq. (5) preserves the same value as **mask**. Thus the forward pass remains unchanged. During the back-propagation, an additional **mask** term is introduced in computing the gradient of $\mathbf{O}$ with respect to the gating threshold $\Delta$ in Eq. (6) because of the element-wise square. Consequently, the update of $\Delta$ only requires the result of $\mathbf{mask} \odot \mathbf{O}_{lb}$ which has already been computed during the forward pass.

$$\frac{\partial \mathbf{O}}{\partial \Delta} \approx \frac{\partial \mathbf{O}}{\partial \mathbf{mask}} \frac{\partial \mathrm{sigmoid}(\alpha(\mathbf{O}_{hb} - \Delta))}{\partial \Delta} = 2 \cdot \mathbf{mask} \odot \mathbf{O}_{lb} \frac{\partial \mathrm{sigmoid}(\alpha(\mathbf{O}_{hb} - \Delta))}{\partial \Delta} \tag{6}$$

The gradient of $\mathbf{O}$ with respect to the weights $\mathbf{W}$ in Eq. (7) employs the same sparse $\mathbf{mask} \odot \mathbf{O}_{lb}$ as the update of $\Delta$. Therefore, precision gating can reduce the computational cost of both forward-progagtion (inference) and back-propagation by the same factor.

$$\frac{\partial \mathbf{O}}{\partial \mathbf{W}} = \frac{\partial \mathbf{O}_{hb}}{\partial \mathbf{W}} + \mathbf{mask}^2 \odot \frac{\partial \mathbf{O}_{lb}}{\partial \mathbf{W}} = \frac{\partial \mathbf{O}_{hb}}{\partial \mathbf{W}} + \frac{\partial \mathbf{mask} \odot \mathbf{O}_{lb}}{\partial \mathbf{W}} \tag{7}$$

### 3.4 OUTLIER CLIPPING

PG predicts important features using low-precision computation. One difficulty with this is that DNN activations are distributed in a bell curve, with most values close to zero and a few large outliers. The top row of Figure 3(a) shows some activations as blue dots, including a single outlier. If we quantize each value to 4 bits (second column) and use 2 most-significant bits in the prediction phase (third column), we see that almost all values are rounded to zero. In this case, PG can only distinguish the importance between the single outlier and the rest of the values no matter what $\Delta$ is. Thus, the presence of large outliers greatly reduces the effectiveness of PG.

To address this, we combine PG with PACT (Choi et al., 2018), which clips each layer's outputs using a learnable clip threshold. The bottom row of Figure 3 shows how clipping limits the dynamic range of activations, making values more uniformly distributed along the quantization grid. Now

the 2 most-significant bits can effectively separate out different groups of values based on magnitude. We apply PACT to PG in CNNs, which commonly use an unbounded activation function such as ReLU. RNNs, on the other hand, typically employ a bounded activation function (e.g., tanh, sigmoid) that often makes PACT unnecessary.

## 4 EXPERIMENTS

We evaluate PG using ResNet-20 (He et al., 2016a) and ShiftNet-20 (Wu et al., 2018a) on CIFAR-10 (Krizhevsky & Hinton, 2009), and ShuffleNet V2 (Ma et al., 2018) on ImageNet (Deng et al., 2009). ResNet is a very popular CNN architecture for image classification. ShiftNet and ShuffleNet are more compact architectures designed specifically for mobile and edge devices. We set the expansion rate of ShiftNet to be 6 and choose the $0.5\times$ variant of ShffleNet V2 for all experiments. On CIFAR-10, the batch size is 128, and the models are trained for 200 epochs. The initial learning rate is 0.1 and decays at epoch 100, 150, 200 by a factor of 0.1 (i.e., multiply learning rate by 0.1). On ImageNet, the batch size is 512 and the models are trained for 120 epochs. The learning rate decays linearly from an initial value of 0.5 to 0.

We also test an LSTM model (Hochreiter & Schmidhuber, 1997) on the Penn Tree Bank (PTB) (Marcus et al., 1993) corpus. The model accuracy is measured by perplexity per word (PPW), where a lower PPW is better. Following the configuration used by He et al. (2016b) and Hubara et al. (2017), the number of hidden units in the LSTM cell is set to 300, and the number of layers is set to 1. We follow the same training setting as described in He et al. (2016b), except that the learning rate decays by a factor of 0.1 at epoch 50 and 90. All experiments are conducted on Tensorflow (Abadi et al., 2016) with NVIDIA GeForce 1080Ti GPUs. We report the top-1 accuracy for all experiments.

We replace the convolutional layers in CNNs and the dense layers in LSTM with the proposed PG block. Moreover, the following hyperparameters in PG need to be tuned appropriately to achieve a low average bitwidth with a high accuracy.

- **The full bitwidth** $B$ – this represents the bitwidth for high-precision computation in PG. $B$ is set to 5 or less on CIFAR-10, 5 or 6 on ImageNet, and 3 or 4 on PTB.
- **The prediction bitwidth** $B_{hb}$ – this represents the bitwidth for low-precision computation.
- **The penalty factor** $\sigma$ – this is the scaling factor of the L2 loss for gating thresholds $\Delta$.
- **The gating target** $\delta$ – the target gating threshold. We use a variety of values $\delta \in [-1.0, 5.0]$ in our experiments.
- **The coefficient** $\alpha$ **in the backward pass** – $\alpha$ controls the magnitude of gradients flowing to $\Delta$. We set $\alpha$ to be 5 across all experiments.

Table 1: **Precision gating (PG) on CNN** – models tested are ShiftNet-20 and ResNet-20 on CIFAR-10, and ShuffleNet V2 $0.5\times$ on ImageNet. We compare PG against uniform quantization (UQ), PACT, and Fix-Threshold. $B_{\text{avg}}$ is the average bitwidth. "fp" denotes floating-point accuracy. "$Sp$" denotes sparsity.

| | Ours | | | | Baselines | | | | | | |
| | **PG** | | | | | UQ | PACT | Fix-Threshold | | | |
| | $B/B_{hb}$ | $Sp$ (%) | $B_{\text{avg}}$ | Acc | Bits | Acc. | Acc. | $B/B_{hb}$ | $Sp$ (%) | $B_{\text{avg}}$ | Acc |
|---|---|---|---|---|---|---|---|---|---|---|---|
| ShiftNet-20 | 5/3 | **55.5** | **3.9** | **89.1** | 8 | 89.1 | 89.0 | 5/3 | 48.8 | 4.0 | 74.3 |
| CIFAR-10 | 5/3 | **96.3** | **3.1** | **88.6** | 4 | 87.3 | 87.5 | 5/3 | 67.8 | 3.6 | 67.0 |
| (fp 89.4%) | 3/1 | **71.9** | **1.6** | **84.5** | 2 | 77.8 | 82.9 | 3/1 | 10.1 | 2.8 | 64.3 |
| ResNet-20 | 4/3 | **78.2** | **3.2** | **91.7** | 8 | 91.6 | 91.2 | 4/3 | 58.7 | 3.4 | 88.3 |
| CIFAR-10 | 3/2 | **90.1** | **2.1** | **91.2** | 4 | 91.1 | 90.9 | 3/2 | 71.0 | 2.3 | 74.2 |
| (fp 91.7%) | 2/1 | **71.5** | **1.3** | **90.6** | 2 | 84.0 | 90.1 | 2/1 | 21.6 | 1.8 | 71.9 |
| ShuffleNet | 6/4 | **57.2** | **4.8** | **59.7** | 8 | 59.1 | 59.1 | 6/4 | 52.6 | 4.9 | 33.6 |
| ImageNet | 6/4 | **62.2** | **4.7** | **59.3** | 6 | 57.8 | 57.1 | 6/4 | 58.5 | 4.8 | 32.7 |
| (fp 59.0%) | 5/3 | **41.9** | **4.1** | **58.0** | 5 | 57.0 | 56.6 | 5/3 | 40.4 | 4.2 | 27.7 |

## 4.1 CNN Results

To compare the hardware execution efficiency across different techniques, we compute the *average bitwidth* ($B_{avg}$) of all features in a DNN model:

$$B_{avg} = B_{hb} + (1 - Sp) \times (B - B_{hb}) \tag{8}$$

Here $Sp$ denotes sparsity in terms of the percentage of low-precision activations (i.e., number of unimportant features divided by total features). The computational cost of DNNs is proportional to the average bitwidth. Our results on CNNs are presented in Tables 1 and 2.

We first compare PG against two widely adopted quantization schemes — uniform quantization (UQ) and PACT (Choi et al., 2018). In Table 1, columns 3 and 4 show the average bitwidth and model accuracy of PG, respectively; columns 5-7 list the bitwidth and corresponding model accuracy of UQ and PACT, respectively. At each row, PG achieves a better accuracy with a lower average bitwidth ($B_{avg}$ vs. Bits). Specifically, PG achieves **6.7%** and **1.6%** higher accuracy with **1.25**$\times$ computational cost reduction, and **6.6%** and **0.5%** higher accuracy with **1.54**$\times$ computational cost reduction than 2-bit UQ and PACT on ShiftNet-20 and ResNet-20 for CIFAR-10 (rows 3 and 6), respectively. We observe the same trend on ShuffleNet for ImageNet, where PG improves the accuracy of 5-bit UQ and PACT by **1.0%** and **1.4%** with **1.22**$\times$ computational cost reduction (row 9). It is also worth noting that PG can recover the accuracy of the floating-point ShiftNet-20, ResNet-20, and ShuffleNet V2 $0.5\times$ with **3.9, 3.2**, and **4.7** average bitwidth, respectively. This demonstrates that PG, using a learnable threshold, can predict unimportant features and reduce their bitwidth without compromising accuracy. The results compared to quantization baselines are visualized in Figure 4, which plots accuracy vs. average bitwidth — uniform quantization (squares), PACT (triangles), and PG (circles) are shown on separate curves. Results closer to the upper-left corner are better.

We then compare PG with Fix-Threshold, which is an extension of the zero-prediction (Lin et al., 2017; Song et al., 2018). Lin et al. (2017); Song et al. (2018) explicitly predict ReLU-induced zeros during inference. To have a fair comparison, we extend their technique to predict an arbitrary fixed threshold to achieve the same or lower $B_{avg}$ as PG, reported as Fix-Threshold in Table 1. For CIFAR-10, we observe that PG achieves **20.2%** and **18.7%** higher accuracy with **1.75**$\times$ and **1.38**$\times$ computational cost reduction than Fix-Threshold on ShiftNet-20 and ResNet-20 (rows 3 and 6), respectively. The gap in accuracy becomes even larger on ShuffleNet V2 for ImageNet dataset. With the same or a lower average bitwidth, the accuracy of PG is at least **26%** higher than Fix-Threshold (rows 7-9). In conclusion, PG consistently outperforms Fix-Threshold because PG uses a learnable gate function which can adjust its threshold to the clipped activation distribution.

Table 2: **Comparison with SeerNet on CNN** – compare PG against SeerNet under similar model prediction accuracy. In SeerNet the average bitwidth $B_{avg} = B_{hb} + (1 - Sp) \times B$.

| | PG | | | | SeerNet | | | |
|---|---|---|---|---|---|---|---|---|
| | $B/B_{hb}$ | $Sp$ (%) | $B_{avg}$ | Acc | $B/B_{hb}$ | $Sp$ (%) | $B_{avg}$ | Acc |
| ShiftNet-20 | 5/3 | **96.3** | **3.1** | **88.6** | 12/8 | 49.7 | 14.0 | 85.4 |
| ResNet-20 | 3/2 | **90.1** | **2.1** | **91.2** | 6/4 | 51.1 | 6.9 | 91.2 |
| ShuffleNet V2 $0.5\times$ | 6/4 | **62.2** | **4.7** | **59.3** | 8/6 | 30.8 | 11.5 | 58.9 |

We further compare PG with SeerNet (Cao et al., 2019) in Table 2. For each convolutional layer, SeerNet executes a quantized version of the same layer to predict the output sparsity. It then computes in full precision the output activations that are not suppressed according to the prediction. Since the code of SeerNet is currently unavailable, we implement the network in Tensorflow and boost its accuracy by retraining the network. We reduce the average bitwidth of SeerNet while keeping a comparable accuracy as PG. For CIFAR-10, PG reduces the computational cost of ResNet-20 **3.3**$\times$ more than SeerNet at the same prediction accuracy. Meanwhile for the hardware-friendly ShiftNet-20, PG achieves **3.2%** higher accuracy with **4.5**$\times$ less compute than SeerNet. For ImageNet, PG on ShuffleNet also achieves **0.4%** higher accuracy with **2.4**$\times$ computational cost reduction than SeerNet. It is worth noting that SeerNet does not reuse the outputs from the quantized layer, which may incur a nontrivial overhead in execution time. In contrast, when PG invokes the update phase, it reuses the outputs of the low-precision computation from the prediction phase.

Table 3: **Sweeping manually set thresholds** – we sweep a series of manually set thresholds for ResNet-20 on CIFAR-10. Compared to manually setting thresholds, PG achieves a better model accuracy (91.2%) with a larger sparsity (90.1%).

| $B/B_{hb}$ | Fix-Threshold | $Sp$ (%) | $B_{avg}$ | Acc |
|---|---|---|---|---|
| 3/2 | 3 | 86.0 | 2.1 | 65.9 |
| 3/2 | 2 | 80.2 | 2.2 | 70.7 |
| 3/2 | 1 | 71.0 | 2.3 | 74.2 |
| 3/2 | 0 | 56.7 | 2.4 | 75.8 |
| 3/2 | -1 | 35.2 | 2.6 | 83.5 |
| 3/2 | -2 | 24.5 | 2.8 | 86.9 |
| 3/2 | -4 | 13.4 | 2.9 | 88.7 |

In order to evaluate the efficacy of the learnable thresholds, we further sweep a series of manually set thresholds on the ResNet-20 for CIFAR-10. Table 3 shows the results where we use a dual-precision mode of $B/B_{hb} = 3/2$. As the threshold decreases from 3 to -4, the average bitwidth in the update phase consistently increases. This is expected because we compute more output features in a high precision. The model prediction accuracy therefore increases. Compared to these manually set thresholds, PG achieves a much improved model accuracy (91.2%) with a higher sparsity (90.1%).

Table 4: **PG with and without sparse back-propagation (SpBP) on CNNs**.

| | PG | | PG w/ SpBP | | | |
|---|---|---|---|---|---|---|
| | $B_{avg}$ | $Sp$ (%) | $B_{avg}$ | $Sp$ (%) | $B/B_{hb}$ | Acc |
| ShiftNet-20 | 4.0 | 49.3 | **3.9** | **55.5** (↑6.2) | 5/3 | 89.1 |
| | 3.3 | 84.0 | **3.1** | **96.3** (↑12.3) | 5/3 | 88.6 |
| ResNet-20 | 3.4 | 58.2 | **3.2** | **78.2** (↑20.0) | 4/3 | 91.7 |
| | 2.2 | 76.5 | **2.1** | **90.1** (↑13.6) | 3/2 | 91.2 |
| ShuffleNet V2 0.5× | 5.3 | 36.6 | **4.8** | **57.2** (↑20.6) | 6/4 | 59.7 |
| | 5.1 | 43.1 | **4.7** | **62.2** (↑19.1) | 6/4 | 59.3 |

To quantify the impact of sparse back-propagation described in Section 3.3, we run PG with and without sparse back-propagation on CNNs. Table 4 compares the sparsity in the update phase of PG with and without back-propagation under the same model accuracy and average bitwidth. Interestingly, we find that the sparsity in the update phase of PG with sparse back-propagation is consistently higher than that of PG without sparse back-propagation across the tested models and datasets. For both CIFAR-10 and ImageNet, the sparsity increases by between 6% and 21%. We hypothesize that sparse back-propagation zeros out the gradients flowing to non-activated LSB convolutions in the update phase, which leads to a higher sparsity.

## 4.2 LSTM RESULTS

PG also works well on RNNs. Table 5 reports our results applying PG on an LSTM model for the PTB corpus. Here, we only compare with the uniform quantization since PACT, Fix-Threshold, and SeerNet do not work for sigmoid or tanh activation functions. Although Hubara et al. (2017) claim that quantizing both weights and activations to 4 bits does not lower PPW, we observe a PPW degradation when $B$ decreases from 8 to 4 bits in our implementation. The LSTM with 8-bit activations is therefore considered as the full-accuracy model. We observe the same trend as in the CNN evaluation where PG enables the 3-bit LSTM cell to improve the PPW by **1.2%** and reduce the computational cost by **2.7×** compared to the 8-bit uniform quantization.

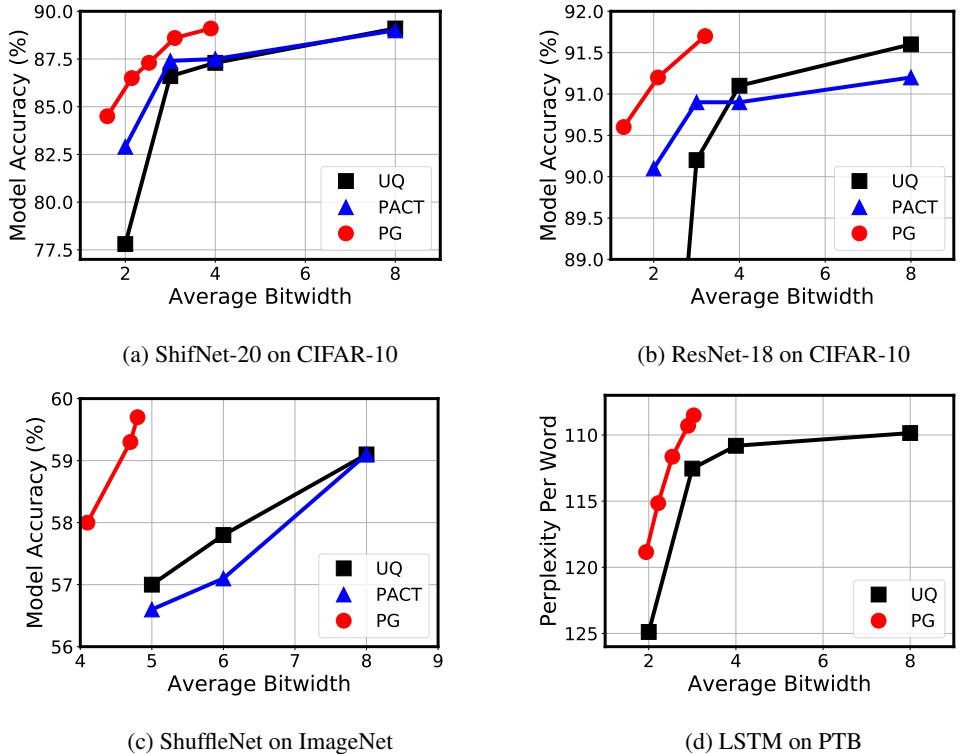

Figure 4: **Precision gating (PG) results on CNNs and LSTM –** compare PG against uniform quantization (UQ) and PACT.

Table 5: **PG on LSTM –** the dataset used is Penn Tree Bank (PTB). The metric is perplexity per word (PPW) and lower is better. Floating-point PPW is 110.1.

| Base | | PG | | | |
|---|---|---|---|---|---|
| Bits | PPW | $B/B_{hb}$ | $Sp\,(\%)$ | $B_{\text{avg}}$ | PPW |
| 8 | 109.8 | 4/2 | 48.4 | **3.0** | **108.5** |
| 4 | 110.8 | 4/2 | 54.9 | **2.9** | **109.3** |
| 2 | 124.9 | 3/1 | 53.0 | **1.9** | **118.8** |

### 4.3 SPARSE KERNEL SPEEDUP

The sparse update phase of PG can be implemented efficiently with a new kernel called sampled dense-dense matrix multiplication (SDDMM). A convolutional layer with PG is then factorized into a regular low-precision convolution bounded with a low-precision SDDMM. To evaluate the potential speedup of PG, we implement the SDDMM kernel in Python leveraging a high performance JIT compiler Numba (Lam et al., 2015) and test it on the ResNet-20 model for CIFAR-10. Table 6 shows the layer-wise sparsity and the kernel speedup compared to the dense matrix multiplication baseline on Intel Xeon Silver 4114 CPU (2.20GHz). With the high sparsity (from 76% to 99%) in each layer induced by PG, the SDDMM kernel achieves up to 8.3× wall clock speedup over the general dense matrix-matrix multiplication (GEMM) kernel. The significant wall clock speedup shows a good potential of deploying PG on commodity hardware.

For a GPU implementation, we need to replace the GEMM kernel with the SDDMM kernel to accelerate the update phase. Mainstream deep learning frameworks such as Tensorflow currently do not provide a built-in operator for SDDMM, which is essential for achieving high performance on GPUs. Nevertheless, Nisa et al. (2018) have recently demonstrated that a highly optimized SDDMM kernel with a similar sparsity shown in Table 6 can achieve about 4× speedup over a GEMM ker-

Table 6: **SDDMM kernel sparsity and speedup** – We report optimized kernel execution time and wall-clock speedup of each layer in ResNet-20 for CIFAR-10.

| Layer ID | 1 | 3 | 5 | 7 | 9 | 11 | 13 | 15 | 17 |
|---|---|---|---|---|---|---|---|---|---|
| $Sp$ | 85% | 94% | 87% | 76% | 98% | 99% | 91% | 98% | 97% |
| Execution Time (ms) | 5.4 | 3.3 | 4.9 | 3.6 | 1.5 | 1.1 | 1.5 | 1.0 | 1.2 |
| Wall Clock Speedup | 3.2× | 5.1× | 3.3× | 2.3× | 6.2× | 8.3× | 3.2× | 6.2× | 5.2× |

nel. This shows strong evidence that PG has a potential to obtain high speedup on GPUs as well. Additionally, our approach is a good fit for the specialized accelerator architectures proposed by Lin et al. (2017) and Song et al. (2018). Due to the high sparsity, DNNs with PG are estimated to get at least 3× speedup and 5× energy efficiency on these dedicated hardware accelerators. We leave the deployment of the SDDMM kernel on GPU and dedicated hardware to future work.

## 5 CONCLUSIONS

We propose precision gating, a dynamic dual-precision quantization method that can effectively reduce the computational cost of DNNs. PG assigns higher precision to important features and lower precision to the remaining features at run-time. The proposed technique is end-to-end trainable, allowing individual models to learn to distinguish important and unimportant features. Experimental results show that PG outperforms state-of-the-art quantization and prediction approaches by a large margin on both CNN and RNN benchmarks on datasets such as Imagenet and Penn Tree Bank. We will release the source code on the author's website.

ACKNOWLEDGMENTS

This work was supported in part by the Semiconductor Research Corporation (SRC) and DARPA. One of the Titan Xp GPUs used for this research was donated by the NVIDIA Corporation. We thank Jordan Dotzel (Cornell) for his helpful discussions during the camera-ready revision.

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

## A    APPENDIX

### A.1    FEATURE VISUALIZATION

In precision gating, we expect that the model will learn to compute features whose prediction values exceed the threshold using a high-precision while keeping others computed using a low precision. In the image recognition task, we expect that high-precision features are mostly in the region where an object lies. To provide evidence that PG can effectively learn to identity those regions, in Figure 5, we visualize the decision maps extracted from the final convolutional layer that is modified to support PG in the ResNet-20 model. A decision map is a gray scale 2D image that has the same spatial size as the output feature map in the same convolutional layer. The brighter a pixel is in the decision map, the more probably the same spatial location in the output feature map will be computed using a high precision. The first row contains the original input images in CIFAR-10, and the second row shows their corresponding decision maps. We can see clearly that in each decision map, the locations of bright pixels roughly align with the object in the original image.

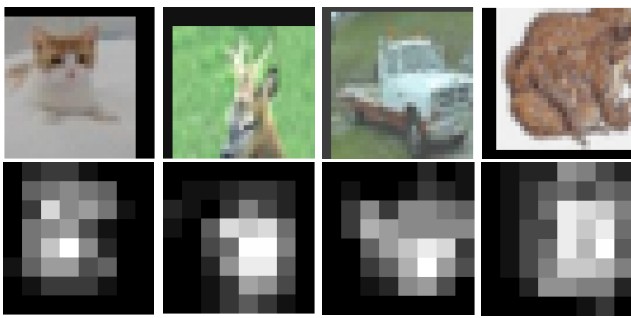

Figure 5: **Visualization of gating ratio –** Top: feature maps from the final precision gating block in ResNet-20 on CIFAR-10. Bottom: ratio of computing using a high-precision (brighter pixel means higher ratio). PG effectively identifies the location of the object of interest and increases bitwidth when computing in this region.

### A.2    ADDITIONAL RESULTS

We provide more supplementary results in this section as shown in Table 7.

Table 7: **Precision gating (PG) on CNN –** additional models tested are ResNet-32 and ResNet-56 on CIFAR-10. We compare PG against uniform quantization (UQ), PACT, and Fix-Threshold. "fp" is floating-point accuracy. "$Sp$" is sparsity.

|  | Ours | | | | Baselines | | | | | | |
|---|---|---|---|---|---|---|---|---|---|---|---|
|  | **Precision Gating** | | | | UQ | PACT | Fix-Threshold | | | |
|  | $B/B_{hb}$ | $Sp$ (%) | $B_{\text{avg}}$ | Acc | Bits | Acc | Acc | $B/B_{hb}$ | $Sp$ (%) | $B_{\text{avg}}$ | Acc |
| ResNet-32 |  |  |  |  | 8 | 92.3 | 91.9 |  |  |  |  |
| (fp 92.4%) | 3/2 | **96.3** | **2.0** | **92.0** | 4 | 92.0 | 91.6 | 3/2 | 94.4 | 2.0 | 45.6 |
| ResNet-56 | 4/3 | **93.0** | **3.1** | **93.0** | 8 | 92.9 | 92.5 | 4/3 | 91.0 | 3.1 | 90.2 |
| CIFAR-10 | 3/2 | **98.2** | **2.0** | **92.5** | 4 | 92.3 | 92.1 | 3/2 | 96.1 | 2.0 | 50.0 |
| (fp 92.9%) | 2/1 | **90.4** | **1.1** | **92.0** | 2 | 88.5 | 91.8 | 2/1 | 86.9 | 1.1 | 14.2 |

