# OpenReview forum: "Precision Gating: Improving Neural Network Efficiency with Dynamic Dual-Precision Activations"
_ICLR.cc/2020/Conference — Accept (Poster)_

### Official Review · AnonReviewer3 · 2019-10-21
**Official Blind Review #3**

**Rating:** 6

**Review:**

This paper presents an interesting quantization technique that is, unusually, end-to-end trainable and not just an inference technique. According to the experiments, the method achieves better performance and computational savings as compared to other quantization method baselines. The results are admirably demonstrated on a variety of models, including CNN and RNN-based neural nets, as well as on several datasets in different domains, including ImageNet, CIFAR10, and PTB. We see the method seems to generalize across all of these.

Nevertheless, while I found this is very interesting work, I have a number of issues with the experiments, which I'll go into below. I feel this work is being released prematurely and could use some more polish to help sell the method better. Below are a few remarks and questions for the authors that would be helpful to be answered.

* Why only report on ResNet-18? It would be far more useful to show numbers against ResNet-50. It would also be useful to show the non-quantized best results on these models and datasets.
* I wish more effort had been spent to analyze the experiments. For example, I am not sure I understand the effects of the threshold on this method. What happens when it's set manually?
* How exactly is computation cost savings calculated so crudely? If it uses B_avg, why not calculate the bitwidth per layer and sum things up? Using B_avg strikes me as being quite crude.
* When the authors address runtime changes except in table 6, they changed their baseline to a vanilla ResNet-18 with dense weights. What are the runtime effects relative to ShuffleNet and ShiftNet?

**Experience Assessment:**

I have published one or two papers in this area.

**Review Assessment: Checking Correctness Of Derivations And Theory:**

N/A

**Review Assessment: Checking Correctness Of Experiments:**

I assessed the sensibility of the experiments.

**Review Assessment: Thoroughness In Paper Reading:**

I read the paper at least twice and used my best judgement in assessing the paper.

---

> ### Author Response · Authors · 2019-11-11
> **Clarifications for the concerned issues**
>
> Thank you for the insightful comments that have led to improvements in our paper. Below are our answers to your questions.
>
> Q1: "Why only report on ResNet-18? It would be far more useful to show numbers against ResNet-50. It would also be useful to show the non-quantized best results on these models and datasets."
>
> Reply: In Table 1, we report the results of the floating point baseline for the three models in the first column as “fp”. For example, the floating-point ShiftNet-20 CIFAR-10 model has an accuracy of 89.4%. In Table 4, we report the floating-point PPW of the LSTM PTB model in the caption.
>
> To empirically show PG works on more models, we apply PG to ResNet-56 and ResNet-32, then train it on the CIFAR-10 dataset. The training settings are the same as described in Section 4. In the latest revision, the additional results are shown in Table 7 in Section A.2.
>
> Compared to the 8-bit PACT baseline, PG achieves 4x computational cost reduction on both models at the same level of prediction accuracy. The sparsity in ResNet-56 (98.2%) and ResNet-32 (96.3%) are higher than that (90.1%) in ResNet-18 for CIFAR-10 dataset. Compared to the fix-threshold baseline, the accuracy of PG increases by 42.5% for ResNet-56 and 46.4% for ResNet-32 with the same computational cost. This increasing accuracy gap empirically shows that PG also works well on larger DNN models.
>
> Q2: " I am not sure I understand the effects of the threshold on this method. What happens when it's set manually?"
>
> Reply: We can consider the threshold in PG as a measurement of the importance of an output feature. If the MSB result $O_{hb}$ exceeds the threshold, it means that the corresponding output feature is important. We will then compute this important feature in high precision. In the bell shaped activation distribution, the larger a threshold is, the less likely an output feature will exceed the threshold, and thus the more output features will be computed using reduced precision. Hence a large threshold is desired to reduce the computational cost. We introduce a threshold loss to make the threshold approach a large value. Moreover, the threshold is also optimized by minimizing the accuracy loss. As a result, the trainable thresholds are learned to jointly minimize the threshold loss and the accuracy loss.
>
> We’ve already compared PG to manually set thresholds. The results of which are shown  in Table 1 under columns of “Fix-Threshold”. We notice that manually set thresholds yield a much lower model accuracy compared to PG at a similar computational cost.
>
> In the latest revision, we also show the results of sweeping a series of manually set thresholds on the ResNet-18 for CIFAR-10 with $B$/$B_{hb}$=3/2 in Table 8 in Section A.2. As the threshold decreases from 3 to -4, the average bitwidth in the update phase consistently increases. This is expected because we compute more output features in high precision. The model prediction accuracy therefore increases. However, compared to the manually set threshold, PG achieves a much better model accuracy (91.2%) with a larger sparsity (90.1%).
>
> Q3: "How exactly is computation cost savings calculated so crudely? If it uses B_avg, why not calculate the bitwidth per layer and sum things up?"
>
> Reply: The average bitwidth is a good indicator for the compute efficiency when we run PG on customized hardware as it reflects the energy consumption per arithmetic operation. Prior art proposed by Song et al. (ISCA’2018) cited in the paper also adopts the same metric of compute efficiency.
>
> We agree that calculating the bitwidth per layer and summing them up is another good metric. However, it is essentially equivalent to the average bitwidth reported in the paper. The average bitwidth is obtained by normalizing the sum of the bitwidth per layer using the total number of features in the network.
>
> Q4: "When the authors address runtime changes except in table 6, they changed their baseline to a vanilla ResNet-18 with dense weights. What are the runtime effects relative to ShuffleNet and ShiftNet?"
>
> Reply: In Section 4.3, we show the kernel speedup of applying PG to the ResNet-18 model on CIFAR-10 dataset. Since PG works by modifying linear layers (i.e., convolutional layers or fully connected layers) instead of the model architectures, it generalizes to other models containing linear layers.

---

### Official Review · AnonReviewer1 · 2019-10-22
**Official Blind Review #1**

**Rating:** 6

**Review:**

This paper introduces Precision Gating, a novel mechanism to quantize neural network activations to reduce the average bitwidth, resulting in networks with fewer bitwise operations. The idea is to have a learnable threshold Delta that determines if an output activation should be computed in high or low precision, determined by the most significant bits of the value. Assuming that high activations are more important, these are computed at higher precision.

I agree that the following three key contributions listed in the paper are (slightly re-formulated):
1. Introducing Precision Gating (PG), the first end-to-end trainable method that enables dual-precision execution of DNNs and is applicable to a wide variety of network architectures.
2. Precision gating enables DNN computation with a better average bitwidth to accuracy tradeoff than other state-ofthe-
art quantization methods. Combined with its lightweight gating logic, PG demonstrates the potential to reduce DNN execution costs in both commodity and dedicated hardware.
3. Unlike prior works that focus only on inference, precision gating achieves the same sparsity during back-propagation as forward propagation, which reduces the computational cost for both passes.

These contributions are novel and experimental evidence is provided for multiple networks and datasets. The paper is well-written and provides insightful figures to showcase the strengths of the present method. Related work is adequately cited. The paper does not contain much theory, but wherever possible equations are provided to illustrate in detail how the method works.

Experimental results are shown for the datasets CIFAR-10 with ResNet-18 and ShiftNet-20, and ImageNet with ShuffleNet V2 0.5x. On both datasets, PG outperforms uniform quantization, PACT, Fix-Threshold and SeerNet in terms of top-1 accuracy and average bitwidth.
What I am missing is information about the variability of results, since there are no error bars. Are the results averaged over multiple trials (if yes how many?), and is there a difference in variance between the methods? I realize that adding standard deviations to all results in the tables might be infeasible, but a qualitative statement would be interesting. In particular, the random initialization of the hb bits could play a bigger role than lb bits.

The two variants of PG, with and without sparse backpropagation are also investigated, showing that sparse backpropagation leads to more sparsity. To show that the resulting lower average bitwidth gained with PG leads to increased performance, the authors implement it in Python (running on CPU) and measure the wall clock time to execute the ResNet-18 model. Speedups $> 1$ are shown for every layer when using PG. Evidence from other papers is cited to argue that similar speedups are expected on GPUs.

Even though at the moment it is unclear to me how statistically significant the results are, and I strongly recommend commenting on this in the paper, I think the idea of PG and the demonstrated benefits make the paper interesting enough to be accepted at ICLR.

I also have a few questions that I could not get completely from the paper:
1. I am a bit confused by what you call features. Fig. 2 shows by example how the method works for an input $I$. Is $I$, a single number, i.e. a single entry of your input vector, or do you mean the complete input vector?
2. Could you give a bit more insight, how you tuned your hyperparameters, especially $\delta$ and $\sigma$?
3. What exactly does e.g. $\delta=-1$ mean? The network ideally should compute at high precision, when the result when only considering the most significant bits is above -1?

From a hardware point of view, the paper focuses on GPU implementations. I would have hoped for a discussion of suitable custom hardware that could support PG most efficiently.


Minor comments that I would be interested in but did not influence my score
- It seems to me that on the top-left image of Fig. 3, one blue circle (the second largest) is too much? First part shows 8 dots, middle and right only seven?
- Can you please cite a source that DNN activations have outliers (Sec. 3.4)?
- You could also define e.g. one $\delta$ and $\Delta$ per layer, couldn't you? Would be interesting to see if  e.g. thinning out the precision over depth is possible / has advantages.

**Experience Assessment:**

I have published one or two papers in this area.

**Review Assessment: Checking Correctness Of Derivations And Theory:**

N/A

**Review Assessment: Checking Correctness Of Experiments:**

I assessed the sensibility of the experiments.

**Review Assessment: Thoroughness In Paper Reading:**

N/A

---

> ### Author Response · Authors · 2019-11-11
> **Clarifications for the concerned issues**
>
> Thank you for being positive towards our paper and providing insightful comments. Below are our answers to your questions.
>
> Q1: "The variability of results."
>
> Reply: To show the variability of the results, we repeat the training for the ResNet-18 model in the 5th row of Table 1 (i.e., $B$/$B_{hb}$=3/2) 10 times. Since the Fix-Threshold scheme is non-trainable (i.e., no variance), we only compare the variability of PG, UQ and PACT.
>
> The results show that the mean of the model accuracy for PG, UQ and PACT are roughly the same as reported in the paper. The variance of the model accuracy of PG, UQ, PACT is 0.041%, 0.083% and 0.048%, respectively. The empirical results suggest that the variability of the accuracy of PG and UQ are at the same level and lower than that of PACT. The variance of  the average bitwidth of PG is 0.00015 bit across all trials.
>
> The random initialization of hb bits does affect the accuracy of the first few iterations. As we train the model for more epochs, the learning process will eventually adjust the weights and gating thresholds to achieve similar accuracy and average bitwidth.
>
> Q2: "Is $I$ a single number, i.e. a single entry of your input vector, or do you mean the complete input vector?"
>
> Reply: Sorry about the confusion. The input feature map $I$ is a third-order tensor. In Fig. 2 we split each element in the input tensor to the MSB part and the LSB part. We will improve this figure in the future revision.
>
> Q3&4: "What exactly does e.g. $\delta = -1$ mean?" "How you tuned your hyperparameters, especially $\delta$ and $\sigma$?"
>
> Reply: The MSB result (i.e., $W*I_{hb}$) is compared against the learnable threshold ($\Delta$) to determine whether we compute the output features in high precision or not. A higher threshold indicates less features are computed in high precision. The $\delta$ here is the target value for the $\Delta$. We add a threshold loss term $\sigma \left \| \Delta-\delta \right \|^2$ into the objective function to let $\Delta$ approach $\delta$ during training, where the $\sigma$ is the penalty factor that balances the threshold loss and the original accuracy loss.
>
> In your example, $\delta$ is -1, meaning that we want to make $\Delta$ be close to -1. The output features are computed at high precision when the MSB result is above $\Delta$. The final value of the thresholds ($\Delta$) are optimized by jointly minimizing the threshold loss and the accuracy loss.
>
> Practically we look at the distribution of the activations and select a $\delta$ in [0, 5] for CNNs and in [-3, 3] for RNNs. We set a small $\sigma$ (typically in the magnitude of 0.001) for the penalty term.
>
> Q5: "I would have hoped for a discussion of suitable custom hardware that could support PG most efficiently."
>
> Reply: PG is a hardware friendly dynamic quantization scheme as it only requires adding extra comparators to implement dual-precision. It can be directly deployed on recent proposed architectures by Song et al. (2018) and Lin et al. (2017). It is estimated to get at least 3x speedup and 5x energy efficiency with the level of sparsity reported for ResNet-18 CIFAR-10. Moreover, there is a line of research on DNN hardware acceleration that proposes to use bit serial / bit parallel execution (Lee et al., UNPU: A 50.6TOPS/W unified deep neural network accelerator with 1b-to-16b fully-variable weight bit-precision, ISSCC’2018) to accelerate ultra low precision networks, which is also a good fit for the precision gating networks. We will add additional discussions on the custom hardware in the future revision.
>
> Q6: "On the top-left image of Fig. 3, one blue circle (the second largest) is too much?"
>
> Reply: Yes, thank you for pointing this out!  The second largest blue circle needs to be deleted. Fig. 3 is fixed in the latest revision.
>
> Q7: "Can you please cite a source that DNN activations have outliers (Sec. 3.4)?"
>
> Reply: (1) Choi et al., PACT: Parameterized Clipping Activation for Quantized Neural Networks, arxiv:1805.06085, May 2018:  It clips the outliers of activations as discussed in section 3.4.
> (2) Park et al., Value-aware Quantization for Training and Inference of Neural Networks, ECCV’2018: This paper discusses the outlier in neural networks and proposes to assign higher precision to these outliers.
> (3) Zhao et al., Improving Neural Network Quantization using Outlier Channel Splitting, ICML’2019: This paper proposes to split the channels to decrease the outlier by half.
>
> We will also add these citations when we revise the paper.
>
> Q8: "You can also define one $\delta$ and $\Delta$ per layer, couldn't you?"
>
> Reply: Right, it’s possible to define a single delta and sigma per layer. This would reduce the hardware cost as we can store fewer thresholds. We plan to run more experiments to understand if this simplification would degrade the accuracy. In this paper we investigated a more general and fine-grained case to show the potential of computing with dual precisions in DNNs.

---

> > ### Comment · AnonReviewer1 · 2019-11-15
> > **Thanks for the clarification!**
> >
> > Thank you to the authors for the clarifications and additional experiments.
> > I agree with the other reviewers that the idea in this paper is a strong contribution to the field.
> > However, I also see that the claims in the paper would be better supported by more experiments and a GPU implementation.
> > Therefore I will keep my score at 6.

---

### Official Review · AnonReviewer2 · 2019-11-05
**Official Blind Review #2**

**Rating:** 6

**Review:**

This paper outlines a new method that allows using a variety of precision in the numerical representation of the network to increase performance (both in terms of accuracy and speed). They learn a threshold value for which all activation values above the threshold are learned at full precision, while all below are learned at reduced precision. This enables substantial performance gains.

The authors summary of the contributions made in the paper is accurate.

The paper is well written and clearly articulates a contribution to the literature. As such, I think it should be accepted. However, the major question I had as I read the paper was the efficacy on GPU, which the paper discusses, but does not implement, nor show any empirical results for, which weakens the paper. Most deep learning happens on GPUs (or similar accelerators), so until this technique is implemented there, it is of limited use. It is still a contribution to the literature, but the paper would be significantly strengthened with a GPU implementation. Additionally, the experimental evidence is lacking. More experiments would also strengthen the paper.

If these changes were made I would change my score to 8 (accept). I do think that the work is slightly premature, and would benefit significantly from adding GPU results and additional experiments. The contribution is strong, however, and should be published in some form, either now, or at a future date.

For the experimental setup, I had a few questions:

1) What happens with fixed thresholds? E.g. doing a sweep over fixed values.

2) How do the results vary for different initialization schemes?

3) How do the results vary with the 5 hyperparameters listed? How were they chosen?

4) How consistent are the results? i.e. what happens if the experiments are repeated N times? Do we see the same values?

In short, I would like to see more experiments. The results are encouraging, but brief. Evaluating on more architectures would strengthen the paper.

Overall questions:

- How well does the 1.2% improvement in perplexity compare to SOTA? Please add context for the numbers reported. It's not at all clear how good of an improvement is seen.
- How do the results change with top-5 accuracy vs top-1 accuracy?


Notes which did not affect the review score:

- There are some typos, e.g. “PG computes most features in a low precision and only a small proportion of important features in a higher precision.” Saying “PG computes most features using reduced precision and only a small proportion of important features using high precision” would be more correct. There are similar typos throughout that I have not listed.
- Tables 3 & 4, and Figure 4, are very cramped and hard to read.
Table 1 and 2 are quite crowded; can you rearrange them so they’re easier to read?



**Experience Assessment:**

I have published one or two papers in this area.

**Review Assessment: Checking Correctness Of Derivations And Theory:**

I assessed the sensibility of the derivations and theory.

**Review Assessment: Checking Correctness Of Experiments:**

I carefully checked the experiments.

**Review Assessment: Thoroughness In Paper Reading:**

I read the paper thoroughly.

---

> ### Author Response · Authors · 2019-11-11
> **Clarifications for the concerned issues**
>
> Thank you for being positive towards our paper and providing insightful comments. Below are our answers to your questions.
>
> Q1: "The efficacy of PG on GPUs."
>
> Reply: There are two main challenges for deploying PG on GPUs: 1) no support for executing extremely low-precision arithmetics on GPUs (below 4 bits); 2) no optimized implementation available for the sampled convolution or sampled matrix-matrix multiplication kernel (SDDMM) in current deep learning frameworks.
>
> However, some recent publications have already shown that it is possible to achieve a high speedup for SDDMM on GPUs if the kernel is carefully optimized (discussed in Section 4.3). Also, customized hardware architectures that support low-precision arithmetics and accelerate the kernels in PG have also been demonstrated by prior works introduced in Section 4.3. All these works show strong evidence that PG has a large potential to work on both commodity and dedicated hardware.
>
> Q2: "What happens with fixed thresholds?"
>
> Reply: To compare the accuracy of fix-threshold and PG, we specifically pick the fixed threshold which achieves a similar average bitwidth $B_{avg}$ to PG.
>
> In the latest revision, we show the results of sweeping a series of fixed thresholds on the ResNet-18 for CIFAR-10 with $B$/$B_{hb}$=3/2 in Table 8 in Section A.2.
>
> As the fixed threshold decreases from 3 to -4, the average bitwidth in the update phase consistently increases. This is expected because we compute more output features in high precision. The model prediction accuracy therefore increases. However, compared to the fixed threshold, PG achieves a much better model accuracy (91.2%) with a larger sparsity (90.1%).
>
> Q3: "How do the results vary for different initialization schemes?"
>
> Reply: Different initialization schemes should not affect the results of PG. Since the distribution of the activations per layer does not change with the initialization scheme, the gating target and learnable thresholds will not be affected either.
>
> Q4: "How do the results vary with the 5 hyperparameters listed? How were they chosen?"
>
> Reply: The full bitwidth B is the bitwidth for high precision computation. We want to lower it down as much as we can. But as we can imagine, the smaller it is, the larger the accuracy degradation will be. Same for the prediction bitwidth $B_{hb}$. We want to compute the predictions at the lowest $B_{hb}$ to reduce the computational cost while maintaining the model accuracy.
>
> The $\delta$ is the target value for the $\Delta$. The smaller it is, generally the more output features will be computed using high precision, which leads to a lower compute saving. In the paper, we select the $\delta$ by looking at the distribution of the activations and choose one in the range of [0, 5] for CNNs and in the range of [-3, 3] for RNNs.
>
>  The $\sigma$ is a penalty factor that balances the threshold loss ($\left \| \Delta-\delta \right \|^2$) and the original accuracy loss. Practically we set a small $\sigma$ (typically in the magnitude of 0.001) to have a relatively small threshold loss compared to the accuracy loss.
>
> During back-propagation, we approximate the step function using a sigmoid like function as discussed in Section 3.2. $\alpha$ controls the steepness of the sigmoid and affects the gradient flow. We choose an appropriate $\alpha$ by grid search.
>
> Q5: "How consistent are the results?"
>
> Reply: To show the variability of the results, we repeat the training for the ResNet-18 model in the 5th row of Table 1 (i.e., $B$/$B_{hb}$=3/2) 10 times. Since the Fix-Threshold scheme is non-trainable (i.e., no variance), we only compare the variability of PG, UQ and PACT.
>
> The results show that the mean of the model accuracy for PG, UQ and PACT are roughly the same as reported in the paper. The variance of the model accuracy of PG, UQ, PACT is 0.041%, 0.083% and 0.048%, respectively. The empirical results suggest that the variability of the accuracy of PG and UQ are at the same level and lower than that of PACT. The variance of the average bitwidth of PG is 0.00015 bit across all trials.
>
> Q6: "I would like to see more experiments."
>
> Reply: To empirically show PG works on more models, we apply PG to ResNet-56 and ResNet-32, then train it on the CIFAR-10 dataset. The training settings are the same as described in Section 4. In the latest revision, the additional results are shown in Table 7 in Section A.2.
>
> Compared to the 8-bit PACT baseline, PG achieves 4x computational cost reduction on both models at the same level of prediction accuracy. The sparsity in ResNet-56 (98.2%) and ResNet-32 (96.3%) are higher than that (90.1%) in ResNet-18 for CIFAR-10 dataset. Compared to the fix-threshold baseline, the accuracy of PG increases by 42.5% for ResNet-56 and 46.4% for ResNet-32 with the same computational cost. This increasing accuracy gap empirically shows that PG also works well on larger DNN models.

---

> > ### Comment · AnonReviewer2 · 2019-11-15
> > **Response to clarifications**
> >
> > Thank you for the clarifications, those sound like excellent additions to the paper.
> >
> > I still think that having a GPU implementation would make this work significantly stronger, and would change my score to an 8 if it were present. Without that, I cannot change my score. While the authors did address that comment, I would still like to see some sort of proof of concept.
> >
> > For all my other concerns, the authors have addressed them satisfactorily, and I think this is a solid contribution. The answer to Q2 in particular is great, and has addressed that concern well.
> >
> > I greatly appreciate the answers to Q5, as variability/reproducibility has been a large concern in the recent ML literature.
> >
> > Thank you for adding additional experiments.

---

> ### Author Response · Authors · 2019-11-11
> **Continued clarifications**
>
> We continue to address the questions here.
>
> Q7: "How well does the 1.2% improvement in perplexity compare to SOTA?"
>
> Reply: There are a large variety of RNN models, but the LSTM network is a popular one and is widely adopted in the literature, for example, by Hubara et al. (2017) and He et al. (2016), as the benchmark. To show the effect of PG on RNNs, we modify the linear layers in the LSTM model to support PG. The baseline floating point result reported in the paper (PPW 110) are at the same level of accuracy as that in the literature (PPW 109). In this work we focus on reducing the compute in a neural network. The 1.2% PPW improvement is an extra benefit obtained while PG achieves a 2.7x compute saving compared to the 8-bit uniform quantization baseline.
>
> Q8: "How do the results change with top-5 accuracy vs top-1 accuracy?"
>
> Reply: In the latest revision, we add the top-5 model accuracy measurements for ResNet-18 on CIFAR-10 and ShuffleNet V2 0.5x on ImageNet, and report them in Table 9 in Section A.2.
>
> The change of the top-5 accuracy aligns well with the top-1 accuracy. The magnitude of the change in the top-5 accuracy, however, is  smaller than the top-1. Especially for the ResNet-18 CIFAR-10 model, the top-5 accuracy has barely changed a little before and after adding PG.
>
> Q9: "There are some typos."
>
> Reply: Thank you for pointing out the typos! We’ll correct them in the future revision.
>
> Q10: "Tables 3 & 4, and Figure 4, are very cramped and hard to read. Table 1 and 2 are quite crowded."
>
> Reply: Thank you for the suggestion. We realize that the tables and Figure 4 are dense and could be hard to read. We’ll reorganize them and make them easier to read.

---

### Decision · Program_Chairs · 2019-12-19

**Decision:**

Accept (Poster)

**Comment:**

The submission proposes an approach to accelerate network training by modifying the precision of individual weights, allowing a substantial speed up without a decrease in model accuracy. The magnitude of the activations determines whether it will be computed at a high or low bitwidth.

The reviewers agreed that the paper should be published given the strong results, though there were some salient concerns which the authors should address in their final revision, such as how the method could be implemented on GPU and what savings could be achieved.

Recommendation is to accept.